# Electroencephalography Neurofeedback Training with Focus on the State of Attention: An Investigation Using Source Localization and Effective Connectivity

**DOI:** 10.3390/s24186056

**Published:** 2024-09-19

**Authors:** Wagner Dias Casagrande, Ester Miyuki Nakamura-Palacios, Anselmo Frizera-Neto

**Affiliations:** 1Department of Electrical Engineering, Federal University of Espírito Santo, Vitoria 29075-910, Brazil; frizera@ieee.org; 2Department of Physiological Sciences, Federal University of Espírito Santo, Vitoria 29040-090, Brazil; emnpalacios@gmail.com

**Keywords:** cortical current density, directed transfer function, electroencephalography, neurofeedback

## Abstract

Identifying brain activity and flow direction can help in monitoring the effectiveness of neurofeedback tasks that aim to treat cognitive deficits. The goal of this study was to compare the neuronal electrical activity of the cortex between individuals from two groups—low and high difficulty—based on a spatial analysis of electroencephalography (EEG) acquired through neurofeedback sessions. These sessions require the subjects to maintain their state of attention when executing a task. EEG data were collected during three neurofeedback sessions for each person, including theta and beta frequencies, followed by a comprehensive preprocessing. The inverse solution based on cortical current density was applied to identify brain regions related to the state of attention. Thereafter, effective connectivity between those regions was estimated using the Directed Transfer Function. The average cortical current density of the high-difficulty group demonstrated that the medial prefrontal, dorsolateral prefrontal, and temporal regions are related to the attentional state. In contrast, the low-difficulty group presented higher current density values in the central regions. Furthermore, for both theta and beta frequencies, for the high-difficulty group, flows left and entered several regions, unlike the low-difficulty group, which presented flows leaving a single region. In this study, we identified which brain regions are related to the state of attention in individuals who perform more demanding tasks (high-difficulty group).

## 1. Introduction

Neurofeedback (NF) is a technique based on the principle of operant learning [1]. It consists of having brain activity recorded and monitored, with the objective of modifying patterns in the brain activity. The patterns to be modified, also known as the NF training protocol, are defined according to the objective to be achieved. Theta/beta ratio (TBR), sensorimotor rhythm (SMR), and slow cortical potential (SCP) protocols have proven to be effective [2]. For instance, the TBR protocol is considered a viable alternative in the treatment of ADHD [2]. TBR (4–7 Hz/12–21 Hz) strives to decrease theta and/or increase beta power in central and frontal brain regions. The authors of [3] demonstrated the possibility of differentiating children with ADHD from the control group by absolute values of theta and theta/beta ratio. Based on the evidence demonstrated on the TBR protocol, it is possible to believe that its analysis can help to find the brain regions related to the attentional state, since in the studies noted that it is possible to differentiate attentional states in individuals with and without attentional disorder. In [4], the midline frontal theta, also known as FmT, was found to be related to sustained attention. Additionally, it was identified that the increase in FmT also represents several other related cognitive events, including working memory.

NF is usually applied with electroencephalography (EEG) [5,6]. The information acquired from the EEG is then used by the subjects themselves so that they can control their own performance [7]. Electroencephalographic neurofeedback (NF-EEG) is an important complementary treatment alternative for a variety of neuropsychological disorders [8]. Among them, there are studies on ADHD [9,10], depression [11,12], schizophrenia [13], addiction [14,15], and post-stroke treatment [16,17]. EEG-NF has also been used to improve and enhance brain functions and abilities in healthy people. In [18], healthy elderly people were exposed to neurofeedback sessions where an improvement in working memory performance was reported. In another study, the authors determined the impact of neurofeedback training on dynamic balance in judo athletes [19].

Mathematical methods applied to EEG data have become a relevant method to explore not only brain frequencies, but also brain regions related to specific cognitive processes (source localization). In addition, they enable the investigation of the connectivity between these regions (effective connectivity). One of the limitations of EEG is its low spatial resolution when used to locate and visualize brain electrical activity [20]. To substantially improve the spatial resolution, it is necessary to solve the so-called *inverse problem* of the EEG. To that end, it is valuable to use regularization techniques and methods such as the Minimum Norm Estimate (MNE) [21] and the Low Resolution Electrical Activity Tomography (LORETA) [22]. In [23], a comparison of brain sources of EEG between men and women during exposure to affection stimuli from music videos was performed. LORETA was used to locate regions specifically involved in these emotional responses. In another study, sources located between obese food-addicted and non-food-addicted individuals were compared [24]. In a more recent study, the MNE method was used to compare differences in electrical activity in a working memory test between encephalitic and control groups [25].

There have also been studies that aimed to analyze effective connectivity. Directed functional connectivity analysis based on EEG source signals has been widely used in the study of resting brain networks [26] and in patients with neurological disorders [27,28].

In this study, the *inverse problem* was solved by applying the cortical current density (CCD) source model [29]. After CCD was applied, the *functional location* of EEG recordings were generated by applying the Minimum Norm Estimation (MNE) method [29]. It is a well-known strategy for estimating current density distributions within the brain. Regarding *effective connectivity*, it can be evaluated as a function of frequency with a mathematical method known as Directed Transfer Function (DTF) [30]. This method provides the flow of relevant information between regions for frequencies of interest.

The combination of NF and serious games has been used in the rehabilitation of multiple cognitive deficits [31,32,33]. Serious games are software that combine a “serious” purpose (other than just entertainment purpose) with the structure of a video game [34]. In [35], the integration of a serious game controlled by EEG was carried out. The goal was to train the attention capacity while detecting the attention level by applying machine learning techniques. Another study compared beta wave brain activity (12 Hz–30 Hz) between 2D and 3D serious games [36]. The authors also describe some benefits of using serious games and NF, such as the change in theta activity, which seems to be a good indicator of concentration.

Some studies in the literature suggest that theta and beta frequencies are related to ADHD and, therefore, indicate a complementary treatment with NF using the theta/beta ratio protocol [2,37]. However, researchers have not yet been able to prove the real effectiveness of the protocol [38]. In the present study, we used both source location and effective connectivity methods to investigate the brain areas where theta and beta frequencies are most active.

To the best of our knowledge, there are few studies that have implemented relevant preprocessing to the data collected from EEG. In the present study, in addition to conventional filtering, we implemented outlier filtering and common-mode noise attenuation.

It can also be observed that most of the NF systems applied in the treatment of neuropsychological disorders related to the attention state present a very simple feedback to the subjects. Therefore, there is a need to increase the quality of feedback, for instance, in the form of serious games, so that there is more engagement and interactions of the participant in the NF session. We developed a game that requires an attentional state from the subject, which provides feedback during the NF sessions [39]. The game consists of a rocket that collects stars for points. If the subject is focused, the speed of the rocket increases, and it collects more stars.

To the best of our knowledge, there are no studies that have investigated the functional localization and effective connectivity in participants of neurofeedback sessions separated into two groups with different levels of difficulty in a game. In this study, the participants were separated into a low-difficulty group and a high-difficulty group, where the game was facilitated for the low-difficulty group. The motivation behind this study was to investigate the neuronal electrical activity of the cortex between individuals from these two groups. This was achieved by carrying out a spatial analysis of the EEG data acquired in NF sessions. During the NF sessions, the participants were required to concentrate in order to finish their task. Therefore, we present an analysis and conclusions on the cortical regions that present a higher oscillatory activity related to the state of attention during those sessions.

We also selected a set of cortical regions of interest for further identification of the effective directional connections in both theta and beta frequencies. This enabled us to understand the connectivity performed during the sessions. The results of the comparative analysis are also presented. This study was restricted to theta and beta frequencies because they are widely used and of great relevance in the treatment of neuropsychological disorders such as ADHD [2]. Distribution graphs of the theta and beta relationships present in the three sessions proposed in the study were also generated and are discussed here.

The contributions of this paper can be summarized as follows:Implementation of a comprehensive and innovative preprocessing method for brain data, including noise attenuation and removal of common-mode noise between electrodes;Development of a system for acquiring and classifying the state of attention and a serious game on the unity platform to immerse users in NF sessions.Investigation and analysis of regions related to the attentional state and the connectivity patterns between them for the proposed NF system;Comparison of results between healthy individuals from two groups, low- and high- difficulty, who performed NF sessions assisted by a serious game with a focus on the state of attention.

This paper is organized as follows. Section 2 describes the EEG system, the acquired data, the data preprocessing, and the mathematical methods applied to identify functional localization and effective connectivity. Section 3 describes the results obtained with the applied methods. Section 4 presents the discussion, analysis and comparison with other studies. Section 5 concludes the paper with an indication of future work.

## 2. Materials and Methods

### 2.1. Participants

Forty healthy subjects were recruited to perform the proposed NF sessions (twenty males and twenty females, with a mean age of 24.71 years, SD ± 3.02). They were divided into two groups of twenty people, which we distinguish by calling the low-difficulty group and the high-difficulty group. Dividing into groups was carried out randomly. The number of participants was selected based on practical considerations, including both the time required for image processing and the availability of volunteers.

All participants gave their written informed consent after receiving an explanation of the steps and purpose of our research. Some exclusion criteria were taken into consideration, such as recent drug or alcohol use, recent head trauma, and previous psychiatric diagnosis.

This work was approved by the Ethical Committee from the Federal University of Espirito Santo (UFES)/Brazil with code number CAAE 19403713.6.0000.5060, and it followed the ethical standards of the Human Experimentation Committee of the University of Espirito Santo, Brasil (UFES).

### 2.2. Data Collection and NF Sessions

Participants in this study were asked to participate in NF sessions. A total of three sessions were carried out with each participant, so that we could analyze the brain modulations and their connectivity throughout the process. In each session, the participant would play a game where they would control the speed of a space rocket to collect the most number of rewards (presented in the form of stars) during five minutes. The rocket’s speed increased or decreased according to the duration for which the subject was able to keep their state of attention. This game was developed specifically for this study and has been previously published [39]. The number of sessions and game time were defined based on an initial analysis and a time that would not leave the subjects mentally tired, since the game demands a great deal of attention.

As described, participants were separated into two groups—low- and high-difficulty. The difference between the two groups relies on the difficulty of the game. For the low-difficulty group, it would be easier to maintain a high game speed, i.e., a lower focus on the rocket would be required to speed it up. On the other hand, for subjects in the high-difficulty group, the game would demand a higher level of attention to maintain a higher speed of the rocket. Participants were not aware of which group they were part of.

During the game, brain data were recorded with the EEG Quick-20 signal capture system (Cognionics, San Diego, CA, USA) [40]. The electrodes on headset Quick-20 were cleaned before each session to maintain consistency. NF training was performed using the 19 derivations of the international standard 10/20 system (FP1, FP2, F3, F4, Fz, F7, F8, C3, C4, Cz, T3, T4, T5, T6, P3, P4, Pz, O1 and O2) with a reference on the left ear.

The data were collected and stored using the Cognionics Acquisition Software (CGX Acquisition v66). EEG was recorded at 500 Hz sampling rate and band-pass-filtered from 0.5 to 100 Hz. Each session took approximately 15 min to complete.

The NF sessions were used in three phases, filtering the EEG signals from 1 to 30 Hz by applying a 2nd-order Butterworth filter. The first and second phases (first 10 min) were used to calibrate the system, especially to extract the features of the signals (theta and beta frequencies, theta/beta ratios) and obtain the model of a Support Vector Machine classifier with radial kernel basis function (SVM-RBF) to classify whether the participant was in an attention or inattention state. For this purpose, Matlab functions were used: fitcsvm() to model the classifier and predict() to predict the class.

In the initial phase, EEG data were collected while the user directed their attention towards a space rocket that appeared and disappeared from the screen every 15 s over a 300-second period. The EEG data corresponding to the space rocket’s appearance or disappearance were categorized as attention state or non-attention state, respectively.

In the second phase, the subject used the system already calibrated in the first phase to test it for a period of 300 s and provide new data linked to the state of attention, which was used to recalibrate the system and, thus, improve the probability of correspondence to each of these mental states (state of attention and non-attention).

Then, in the third and final phase, 5 min was used for the NF training. In the neurofeedback training stage, to ensure communication between the user and the feedback given by the system, a Matlab library known as a Lab Streaming Layer (LSL) was used. The LSL consists of a protocol with libraries and tools that allow for the synchronization and transfer of neurophysiological data and biosignals in real time [41]. Then, a matrix buffer was built with 1000 samples per EEG channel, and the feedback update rate was based on 500 samples per second. In our study, we used the EEG data from this last phase to analyze source localization and effective connectivity to understand the regions related to the attentional state.

Participants were instructed to minimize blinking episodes and eye movements, have a relaxed jaw, and minimize body movements during the experiment. All recordings and sessions were performed in a room with a prepared environment in the research laboratory. Lighting and temperature were kept constant during the experiment.

All EEG data collected were processed with special attention to the frontal and temporal leads because these are regions directly related to the mental state of attention [42].

EEG data from all previously mentioned electrodes were stored continuously during sessions. In this sense, data used to validate the system can also be analyzed later.

### 2.3. Data Preprocessing

As mentioned previously, the data collected in the third phase were used to perform the analysis of source localization and effective connectivity to understand the regions related to the attentional state. For this purpose, a preprocessing of the collected signals was performed to generate the images of source localization and effective connectivity.

Initially, the signals were filtered with a Finite Impulse Response (FIR) bandpass filter from 1 to 45 Hz, and they were subsequently filtered to select the frequencies of interest: theta (4–7 Hz) and beta (12–30 Hz). Then, the signals were filtered with a common average reference (CAR) to reject the spatial common interference. In this filtering, the average signal (interference) of all channels involved is calculated, and this signal is subtracted from each channel, thus eliminating spatial interference.

In addition, an outlier filtering was implemented based on the suppression threshold called *gating* [43]. If the amplitude of the EEG signal exceeds a certain threshold, both in the positive or negative directions, it is flagged as an outlier and replaced by the average of the previous 500 samples of the signal. The advantages of this method are the ease of implementation and fast execution, which allow for its use in real time. However, EEG information is inevitably lost when discarding the segment that is contaminated by the outlier. To identify where the contaminated periods were, the mean absolute value (MAV) was used to obtain the characteristic of the EEG signal. Then, K-Means was applied to cluster and define the minimum and maximum thresholds. K-Means clustering is an unsupervised algorithm that aims to cluster a given dataset into different subsets based on some criteria. It maximizes the similarities between the patterns in the same cluster and minimizes them between different clusters [44]. In this study, the K-means method was used for every 500 analysis samples. A sliding window of 100 samples and K = 3 (number of clusters) were used. The minimum and maximum limits were defined by the separation of the data into 3 clusters.

The cortical activity and effective connectivity were estimated using an open source MATLAB software library [45] with graphical user interfaces known as eConnectome (NIITRC—NeuroImaging Tools and Resources Collaboratory) (Software package for imaging brain functional connectivity from electrophysiological signals—http://econnectome.umn.edu/—accessed on 10 May 2023). Compared to other software packages, this one proved to be a flexible platform by allowing changes in the source code, in addition to it providing images of effective brain connectivity.

### 2.4. Functional Localization

Some algorithms were applied to estimate cortical activity. Firstly, the cortical current density (CCD) was applied to solve the EEG inverse problem. This technique improves the spatial resolution of the EEG. Then, along with the CCD model, a set of algorithms, including minimum norm, was used for the image cortical sources. Finally, the Boundary Element Method (BEM) model was used to reconstruct the cortical sources for an individual anatomy. Combining these methods enables us to identify the brain regions that present stronger activation during the NF sessions.

*Cortical current density* is a linear method that consists of estimating cortical activity from EEG signals [46]. For better computational efficiency, the model assumes cortical activity in the form of 7850 dipoles that are positioned perpendicular to the cortical surface with a fixed orientation. The cortex model was constructed based on the standard Montreal Neurological Institute (MNI) brain [47]. Mathematically, the CCD activity at a time t can be estimated through the following formula:(1)CCDactivity(t)=G∗b(t)
where G is a matrix that represents the inverse of the direct model simulating the electrical propagation properties of cortical sources, and b(t) is a linear transformation of the EEG surface potential at a time t. A linear projection of the EEG potential onto CCD vortices tends to reveal EEG signals, resulting in an improved spatial resolution and a better signal-to-noise ratio. Consequently, the calculated classifiers can improve brain activity recognition performance.

### 2.5. Effective Connectivity Analysis

Images of effective connectivity of theta and beta frequencies were generated together since images of brain regions activated at each moment alone do not contain information about how these regions communicate with each other. Effective connectivity images can contribute to the identification of connection patterns between regions related to the mental state of attention.

The cortical Regions of Interest (ROI) to generate effective connectivity images were based on Brodmann Areas, with the following areas being manually selected: anterior prefrontal cortex (10), dorsolateral prefrontal cortex (46), prefronta dorsolateral cortex (9), temporal gyrus (20,21,22), somatosensory association cortex (7), associative visual cortex (19), and primary and secondary visual cortices (17,18). The selected areas were based on the results of the source localization maps.

The directed transfer function (DTF) was used to calculate the effective connectivity between the ROIs. This method consists of a measure based on the theory of Granger causality. This concept aims to describe the influence of one channel on another at a given frequency in a causal form [48]. For this principle to be valid, there must be no other channels influencing the process. So, to treat the multivariate structure of a process with a given number of channels, a multichannel autoregressive model (MVAR) must be taken into consideration. The model is inserted into all channels simultaneously.

According to the Granger causality principle, we can assume that from a weighted sum of *p* previous samples, we can predict the data samples in *k* channels at a time *t*. In addition to the weighted sum, a prediction error E(t) must be added:(2)$$X\left(t\right)=\sum_{j=1}^{p}A\left(j\right)X\left(t-j\right)+E\left(t\right)$$

The X(t) component represents a vector with the data values at a time *t*. A(j) represent arrays known as model parameters. These matrices have size *k* × *k*. And the parameter *p* represents the model order.

When performing the Fourier transform of the MVAR model, the model starts to represent a linear filter H with the presence of noise E and an output signal X:(3)$$H\left(f\right)=\left(\sum_{m=0}^{p}A\left(m\right)exp\left(-2\pi \mathrm{imf}\Delta t\right)\right)$$
(4)$$X\left(f\right)=A^{-1}\left(f\right)E\left(f\right)=H\left(f\right)E\left(f\right)$$

As a result, we have the components X(t), A(t), and E(t) represented in the frequency domain and a matrix H(f) that represents the transfer matrix.

In the frequency domain, the normalized directed transfer function can be defined as follows:(5)$$\gamma_{ij}^{2}\left(f\right)=\frac{{\left|H_{ij(f)}\right|}^{2}}{\sum_{m=1}^{k}{\left|H_{im(f)}\right|}^{2}}$$

The components of the transfer matrix H are represented by $H_{ij(f)}$. The causal influence of the channel *j* on the channel *i* at a given frequency *f* is represented by the term $\gamma_{ij}^{2}\left(f\right)$. Equation (Equation 5) produces a ratio between the influence of channel *j* to channel *i* for all influences to channel *i* when assuming values from 0 to 1.

A major advantage of DTF compared to other methods for estimating functional relationships between biological signals is that it provides an estimate of the causal links between the analyzed signals in terms of strength and direction, which is not possible with conventional methods [49].

The strength of effective directional connectivity, as a function of frequency, was evaluated using mathematical tools included in the eConnectome library, applied to the estimated CCD signals. Statistical significance is reported using the surrogate data method [50,51]. In this work, the surrogate data method known as random phase surrogates was used [50]. It presents the following operations: Fourier transform (FT) of a time series to keep the magnitudes of the Fourier coefficients unchanged; the phases of the fourier coefficients are shuffled randomly to create new replacement data; the inverse FT is performed in the time domain; a surrogate dataset is created. Once this is performed, a new MVAR model is created and fitted to the surrogate data set and, similarly, the DTF values are estimated from the model. This procedure is repeated by shuffling each set of source time series 5000 times, and an empirical distribution is created for the DTF values under the condition that the null hypothesis of no causality is true. Using this distribution, the statistical significance of the DTF value evaluated from the real source time series is evaluated. The significance level used was 0.05.

### 2.6. Statistical Analysis

To identify whether there was any statistically significant difference between the study groups and/or between sessions, we applied one-way ANOVA. We assessed the source location of each region between the high-difficulty and low-difficulty groups for each of the three sessions. We assessed the source location of each region across the three sessions in each study group. And we assessed the regions between sessions at the moments of the theta/beta attention relationship. The level of statistical significance adopted was *p* = 0.05.

## 3. Results

Figure 1 shows the proposed system to generate the functional localization and effective connectivity estimates.

For each participant, theta (4–7 Hz) and beta (12–30 Hz) frequencies were selected to estimate cortical activity and effective connectivity. To facilitate the analyses, images of the average functional location and effective connectivity of each group were generated. The NF sessions were all based on the theta/beta attention training protocol, where the intention is to increase the power of the beta frequency range and reduce theta.

Figure 2 and Figure 3 show the average cortical current density estimates for subjects in the low-difficulty group and the high-difficulty group during the three sessions (S1, S2, and S3). It is possible to identify in the images the regions that showed greater activation (higher normalized power of the cortical current density distribution) for theta and beta frequencies, respectively. The color bar is defined by the maximum absolute value between 0 and 1.

Figure 4 and Figure 5 present the average estimates of effective connectivity for theta and beta frequencies, in addition to the inputs and outputs of the flows. Due to the selected significance level, only statistically significant connectivity modifications are displayed. Connectivity between regions are represented by directional arrows. These arrows indicate the flow of causal information from one given channel to another. The intensity of the flows is represented by the color and width of the arrows. In addition to the arrows, we also have the representation of the outputs and inputs of the flows. The output flow of each channel can be understood as the sum of information from a given channel to all other channels. Its magnitude is coded by the color and size of the sphere drawn in the channel. Likewise, the input of each channel counting the total input of all other channels is visualized by a sphere with the magnitude represented by the sphere’s size.

To complement and deepen the results regarding the theta/beta relationship, the relationships were extracted during moments of attention and inattention in the three sessions proposed to the participants. Boxplot graphs were generated to identify the distribution of these relationships and reach conclusions about the relationships present in each brain region studied.

Figure 6 and Figure 7 demonstrate the distribution under analysis, with Figure 6 representing the participants in the high-difficulty group and Figure 7 representing the participants in the low-difficulty group, respectively. For each of the regions under analysis, six boxplots were plotted, with the first three representing the theta/beta ratio in the first, second, and third sessions, respectively, during the moments of required attention and the other three boxplots representing the first, second, and third sessions, respectively, without required attention.

Table 1, Table 2 and Table 3 present the values obtained through one-way ANOVA that assessed whether there was a statistical difference from the following points of view:The source location of each region between the high-difficulty and low-difficulty groups for each of the three sessions;The source location of each region across the three sessions in each study group;The regions between sessions at the moments of the theta/beta attention relationship.

The following regions were analyzed: LF—left frontal; RF—right front; LDP—left dorsolateral prefrontal; RDP—right dorsolateral prefrontal; LMF—left medial frontal; RMF—right medial frontal; LT—left temporal; RT—right temporal.

## 4. Discussion

This section is divided into subsections to discuss the results obtained from each of the methods applied in the study.

### 4.1. Functional Location of Oscillatory Activity for the Developed Game

In Figure 2, it can be observed that the average result of the high-difficulty group presented high values of cortical current density distribution in the left and right medial prefrontal, dorsolateral prefrontal, and temporal areas. In contrast, subjects in the low-difficulty group showed oscillations in the theta band in the central left and right regions.

Theta activity showed high current density values in the medial prefrontal gyrus during NF sessions in the high-difficulty group. This theta activity seems to correspond to the FmT, a distinct theta activity in the frontal midline area appearing during a concentrated performance, which has been related to the state of focused attention [4].

For the beta frequency range, Figure 3 demonstrates that the region that stood out was the right and left temporal and medial prefrontal region, which was present in subjects of the high-difficulty group. The temporal lobe, as part of the limbic system (amygdala and hippocampus), is believed to play a role in the process of focusing on a task and acting quickly in the presence of distracting stimuli [52]. Moreover, it has been long believed that the temporal area plays an important role in recognition memory and visual, auditory, and language processing [53].

The subjects in the low-difficulty group also presented high values of cortical current density in the central regions for beta frequency. Except the activity in the left dorsolateral prefrontal region that appeared only in the average of the first session, suggesting that it is the first time the user encounters the game, it is necessary to recruit regions related to the state of attention.

The occipital activation seen in the figures were expected, as the training requires visual interaction.

A one-way repeated measures ANOVA showed a significant main effect for the comparison of source location between the two study groups (high-difficulty group vs. low-difficulty groups), both for theta and beta frequencies. For theta frequency, the following significant differences were found: in the second session for the right dorsolateral prefrontal region (F(1,38) = 4.09, F = 5.540) and right temporal region (F(1,38) = 4.09, F = 8.799); in the third session for the right temporal region (F(1,38) = 4.09, F = 4.366). For beta frequency, the following significant differences were found: in the first session for the right temporal region (F(1,38) = 4.09, F = 43.928); in the second session for the left dorsolateral prefrontal (F(1,38) = 4.09, F = 5.926) and right temporal (F(1,38) = 4.09, F = 39.580) regions; in the third session for the left frontal (F(1,38) = 4.09, F = 4.191), left dorsolateral prefrontal (F(1,38) = 4.09, F = 14.776), left medial frontal (F(1,38) = 4.09, F = 5.106), and right temporal (F(1,38) = 4.09, F = 56.962) regions.

A one-way repeated measures ANOVA showed a significant main effect for the comparison of source location between the three study sessions (S1 vs. S2 vs. S3), only for the low-difficulty group. More specifically, for theta frequency, the following regions showed significant changes: left dorsolateral prefrontal (F(2,57) = 3.158, F = 6.271) and right dorsolateral prefrontal (F(2,57) = 3.158, F = 3.198); for beta frequency: left temporal region (F(2,57) = 3.158, F = 4.157).

### 4.2. Effective Connectivity

The connections realized during the game task showed a complex pattern, with many significant increases and decreases in the flow of information. It should be noted that for the theta frequency range (Figure 4), the flows for the high-difficulty group leave and enter several regions, such as the dorsolateral prefrontal region to the frontal, temporal, and parietal regions in the first session. For the low-difficulty group, it was identified that all flows leave a single specific region, as shown in the average of the each session. The beta frequency range (Figure 5) presented the same characteristics as theta frequency.

Oscillatory theta activity was identified in the parietal area for both groups. According to the literature, this area is directly related to calculation tasks [54]. This equal behavior of oscillation in the two groups was expected, since both groups are stimulated by playing the game only with different difficulty levels.

Note that the EEG-based connectivity methods used in this study not only complement the connectivity methods commonly used in Functional Magnetic Resonance Imaging (fMRI) studies, but actually provide additional spectral information.

Another important point to note is that estimating the connectivity in the scalp channels, although there is no technical difficulty, presents interpretation problems. This is due to the high probability of finding spurious associations between the channels as a result of the volume conduction that stains the brain activity through the channels. For this reason, MVAR favors signals that better reflect the activation of individual brain sources or areas of origin. This is one of the advantages of using the eConnectome library.

### 4.3. Theta/Beta Ratio Distribution

With the graphs shown in Figure 6 and Figure 7, we can analyze the distribution of theta/beta ratios in the study regions from the point of view of position, dispersion, and symmetry. Regarding position, it can be observed, as expected, that the median of the theta/beta ratio is lower in moments of required attention and higher in moments of non-required attention, either due to an increase in beta frequency in the region or a decrease in theta frequency in the region (training objective) for both study groups. However, for the high-difficulty group, the difference between the median of the moments of required and non-required attention is greater, proving a greater intensity of training.

Another analysis is in relation to which regions had the greatest difference between moments of attention and non-attention. And it can be observed that the frontal and medial frontal regions (LF, RF, LMF, RMF) are the ones that obtained the greatest differences, demonstrating that they are regions with a stronger relationship with the attentional state.

Another observation is that in all three sessions, users managed to maintain low theta/beta ratio values, often lower than the values from previous sessions.

Regarding dispersion, we can observe that the participants, regardless of the study group, obtained close and well-defined maximum and minimum values, as shown in the graphs.

Regarding symmetry, even with the presence of outliers, the relationships demonstrate a symmetrical distribution, since the medians represented by ‘x’ are very close to the centers of the boxplots in all sessions.

The points present in the graph are considered outliers present in the distribution.

A one-way repeated measures ANOVA showed a significant main effect for the comparison of theta/beta ratios between the three study sessions (S1 vs. S2 vs. S3), only for the high-difficulty group. The regions that showed significant differences were as follows: right dorsolateral prefrontal (F(2,57) = 3.158, F = 3.225) and left temporal (F(2,57) = 3.158, F = 3.731), demonstrating that the game difficulty level influences the theta/beta ratio of the mentioned regions.

## 5. Conclusions

This study analyzes the regions of brain activation and the effective connectivity between them when adult individuals were exposed to NF training with a focus on the state of attention. NF has been shown to be an effective complementary treatment for certain types of cognitive disorders such as ADHD.

A source location method (CCD) was applied to identify the source of the attention state in the theta and beta frequency ranges. This enables the definition of cortical regions of interest to estimate functional connectivity through methods such as multivariate autoregressive (MVAR) modeling and Directed Transfer Function.

Our results show that the frontal regions (particularly the left and right medial prefrontal regions, left and right dorsolateral prefrontal regions) and the temporal lobe (particularly the right temporal lobe) are essential to the task that the subjects were submitted to. However, it is unclear whether this result is specific to the task at hand or general to attentional/cognitive processing. In addition, the analysis of the flow of neurophysiological information showed that subjects in the low-difficulty group presented flows, leaving specific and unique points compared to subjects in the high-difficulty group who presented flows leaving a greater number of areas.

We identified that, from the cortical source estimate and effective connectivity, certain brain regions are related to the state of attention, such as the medial prefrontal regions (right and left), dorsolateral prefrontal (right and left), and right temporal.

This study enables the identification of areas related to the state of attention and the patterns of connectivity between them. We also carried out a comparison between the results of two groups of healthy individuals, split into two groups with different levels of difficulty, with the main focus being the state of attention.

One-way repeated measures ANOVA was used to identify significant changes between groups and study sessions, validating the results discussed.

In addition to the results and validations presented, this work contributes to the development of a system for acquiring and classifying mental attention states, as well as a serious game designed to immerse users in neurofeedback sessions, supporting future research in this area.

Furthermore, boxplot graphs were generated to complement the study and analysis of the regions of interest, where it was possible to observe that the frontal and medial frontal regions are directly related to attentional demand, complementing and proving the results obtained through the cortical source estimation and effective connectivity. The ability of the participants to maintain low theta/beta ratio values between sessions was also identified, proving the efficiency of the training. The idea is that in the long term, as the sessions progress, the values of this ratio become lower.

The results presented motivate further study. The localized electrical neuronal activity and directional information flow are important tools to describe and understand the mental state of attention. This is an important area of study for the investigation of cognitive disorders. We will include even more participants, including individuals with ADHD, so that we can make a comparison with a control group. Other functional connectivity estimation methods will be implemented in order to compare them with the results achieved in this study.

## Figures and Tables

**Figure 1 sensors-24-06056-f001:**
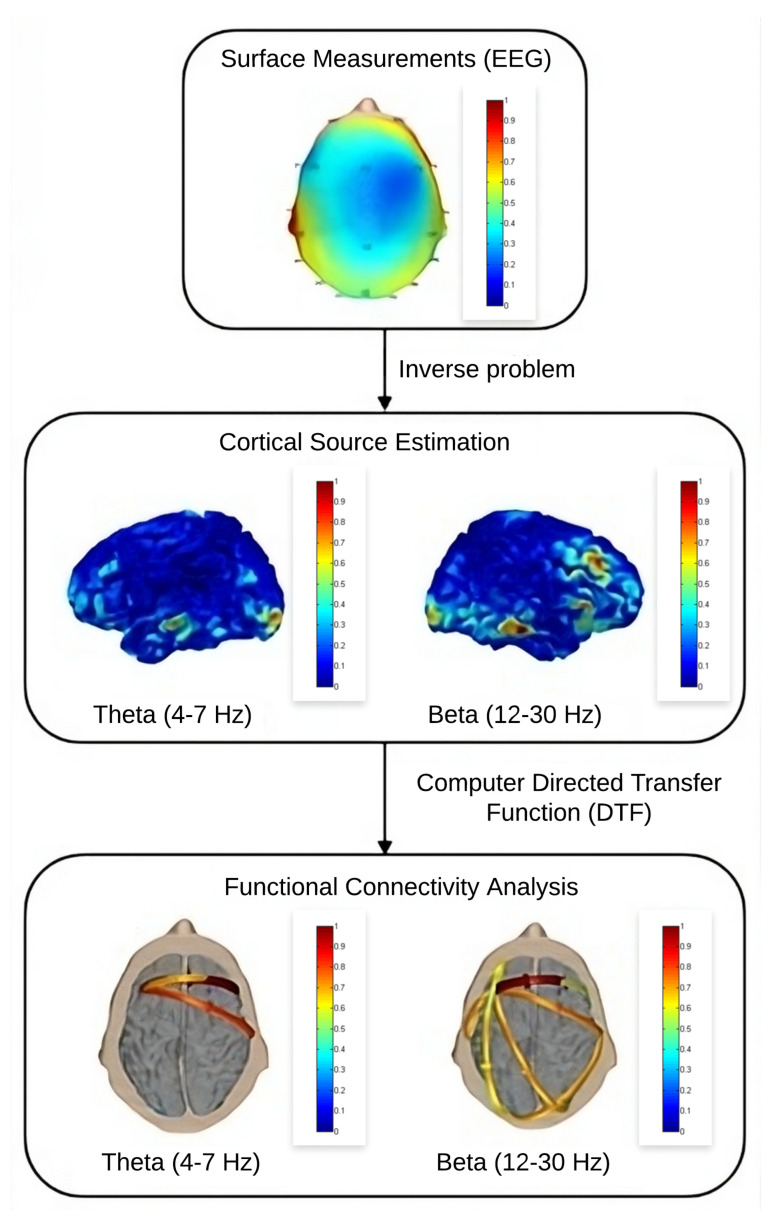
Diagram representing the steps that were followed in the eConnectome library to estimate the cortical activity and effective connectivity of subjects during the applied sessions.

**Figure 2 sensors-24-06056-f002:**
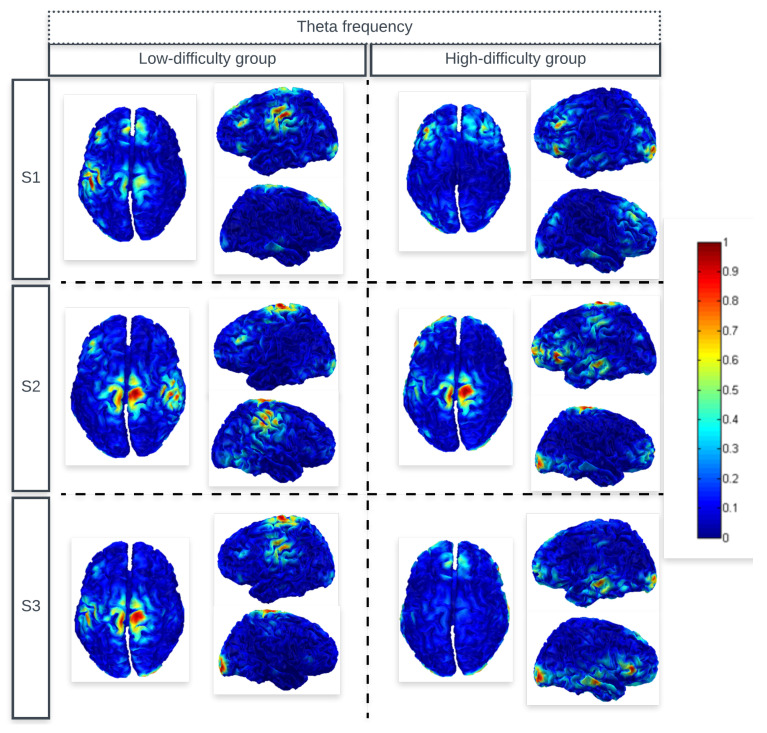
The average cortical current density estimates of both the low-difficulty group and the high-difficulty group subjects for theta frequency.

**Figure 3 sensors-24-06056-f003:**
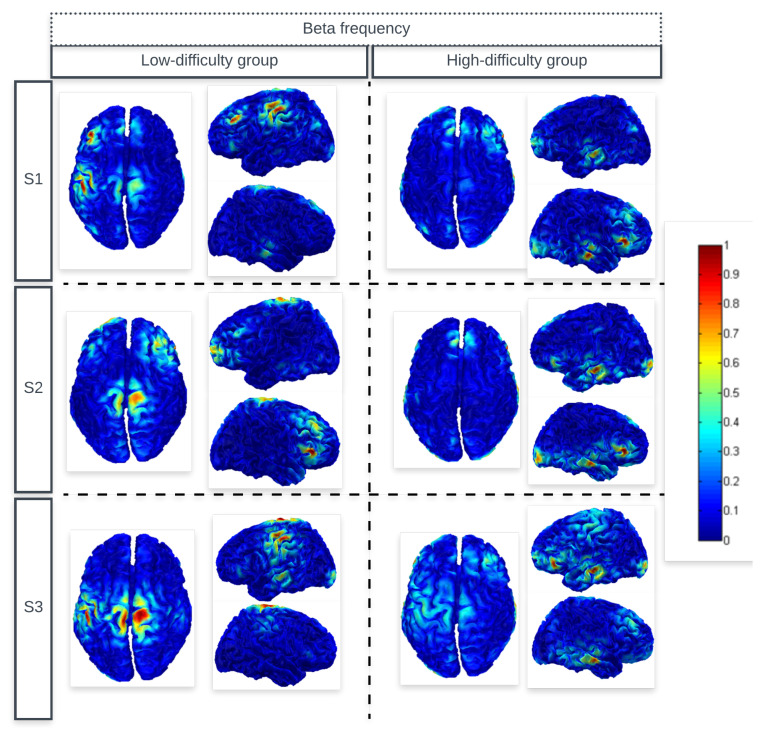
The average cortical current density estimates of both the low-difficulty group and the high-difficulty group subjects for beta frequency.

**Figure 4 sensors-24-06056-f004:**
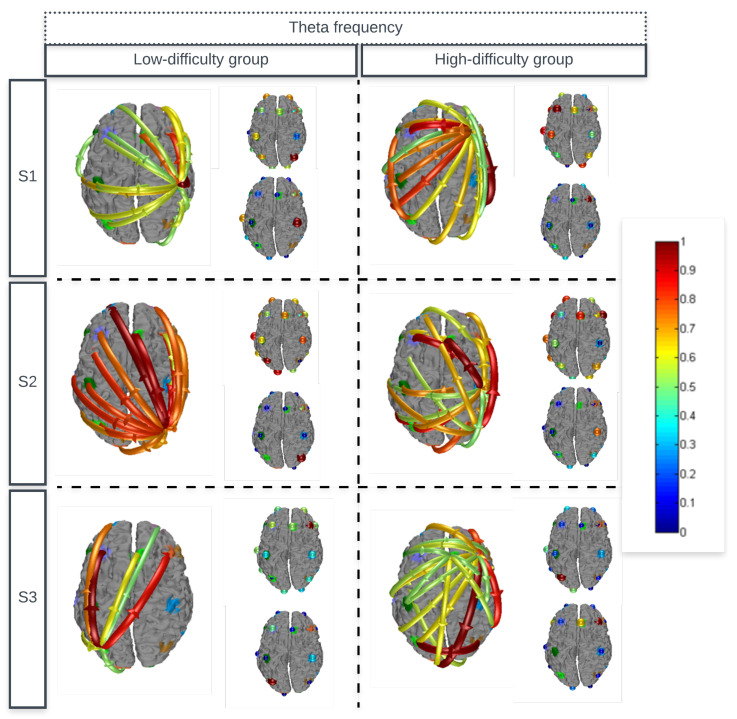
Average estimates of the effective connectivity of the high-difficulty and low-difficulty groups for theta frequency, in addition to the inputs and outputs of the flows.

**Figure 5 sensors-24-06056-f005:**
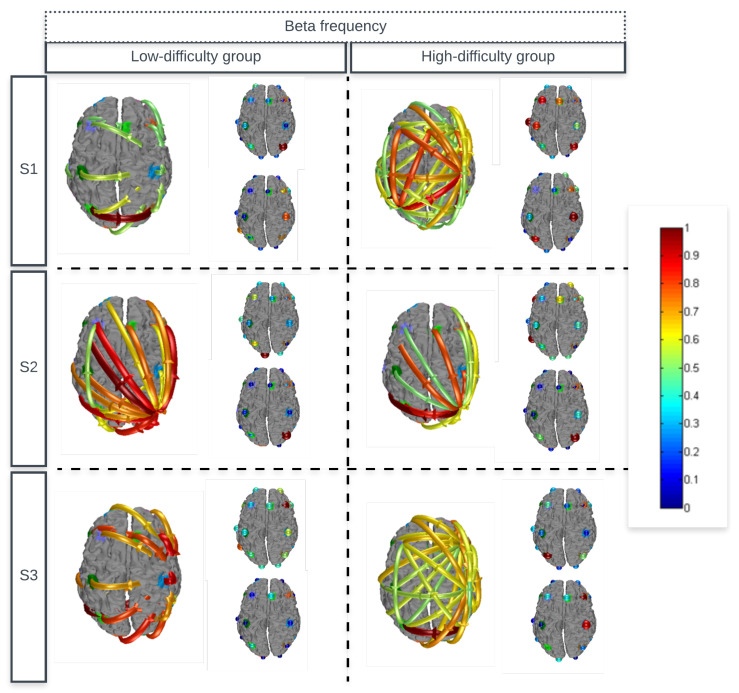
Average estimates of the effective connectivity of the high-difficulty and low-difficulty groups for beta frequency, in addition to the inputs and outputs of the flows.

**Figure 6 sensors-24-06056-f006:**
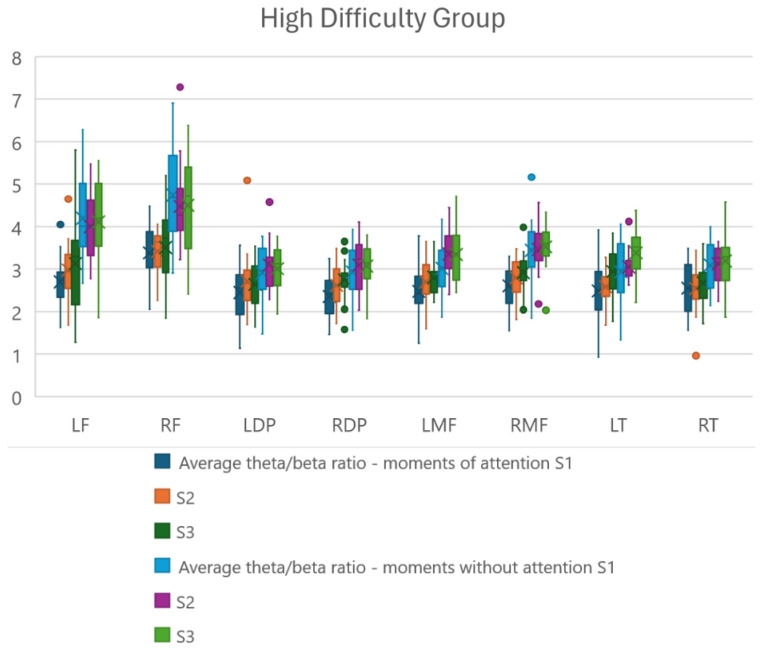
Distribution of the theta/beta ratio for participants in the high-difficulty group during the three training sessions at the moments of attention and inattention in the regions of interest. LF—left frontal; RF—right frontal; LDP—left dorsolateral prefrontal; RDP—right dorsolateral prefrontal; LMF—left medial frontal; RMF—right medial frontal; LT—left temporal; RT—right temporal.

**Figure 7 sensors-24-06056-f007:**
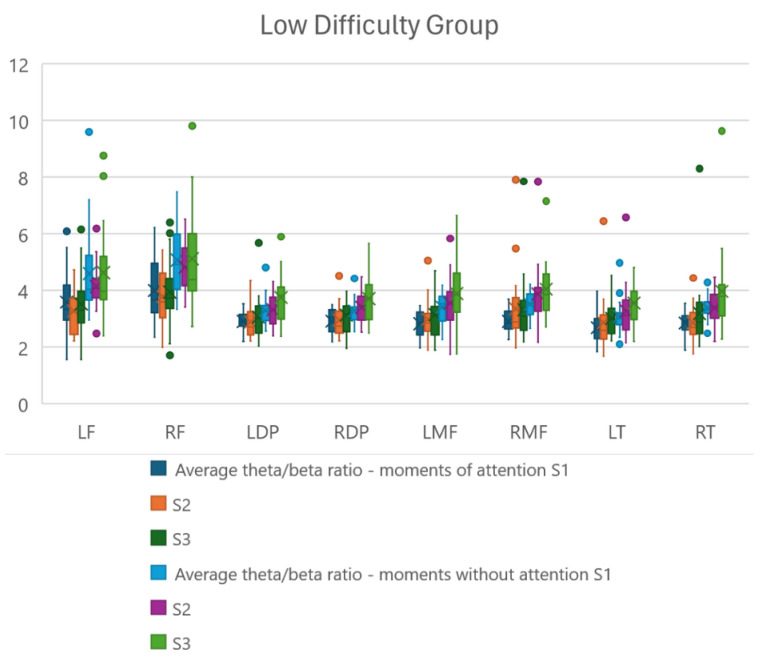
Distribution of the theta/beta ratio for participants in the low-difficulty group during the three training sessions at the moments of attention and inattention in the regions of interest. LF—left frontal; RF—right frontal; LDP—left dorsolateral prefrontal; RDP—right dorsolateral prefrontal; LMF—left medial frontal; RMF—right medial frontal; LT—left temporal; RT—right temporal.

**Table 1 sensors-24-06056-t001:** The source location of each region between the high-difficulty and low-difficulty groups for each of the three sessions.

Theta				Beta			
	**S1**	**S2**	**S3**		**S1**	**S2**	**S3**
**LF**	0.04	1.773	0.297	**LF**	0.317	1.725	**4.191**
**RF**	0.075	0.088	0.259	**RF**	0.286	0.039	0.172
**LDP**	0.811	1.640	3.096	**LDP**	2.472	**5.926**	**14.776**
**RDP**	0.734	**5.540**	0.080	**RDP**	0.455	2.815	0.559
**LMF**	0.244	2.282	0.208	**LMF**	0.105	2.500	**5.106**
**RMF**	0.989	0.761	1.760	**RMF**	0.017	0.611	0.617
**LT**	1.202	0.001	1.294	**LT**	0.760	0.371	1.001
**RT**	0.189	**8.799**	**4.366**	**RT**	**43.928**	**39.580**	**56.962**

**Table 2 sensors-24-06056-t002:** The source location of each region across the three sessions in each study group.

Low-Difficulty Group			High-Difficulty Group		
	**Theta**	**Beta**		**Theta**	**Beta**
**LF**	0.056	1.797	**LF**	0.893	1.534
**RF**	0.740	0.080	**RF**	0.180	0.223
**LDP**	**6.271**	0.593	**LDP**	0.053	0.535
**RDP**	**3.198**	0.286	**RDP**	0.375	2.043
**LMF**	0.562	1.870	**LMF**	1.020	2.241
**RMF**	0.794	0.870	**RMF**	0.455	0.420
**LT**	0.384	**4.157**	**LT**	0.387	0.897
**RT**	0.452	0.240	**RT**	1.639	0.478

**Table 3 sensors-24-06056-t003:** The regions between sessions at the moments of the theta/beta attention relationship.

Theta/Beta Ratio		
	**High-Difficulty Group**	**Low-Difficulty Group**
**LF**	0.150	0.460
**RF**	0.155	0.192
**LDP**	1.069	0.513
**RDP**	**3.225**	0.695
**LMF**	1.811	0.993
**RMF**	2.503	1.070
**LT**	**3.731**	0.956
**RT**	0.406	0.938

## Data Availability

The matlab program developed for this study is open source, with the code available at https://github.com/Wagnerdcasag/Neurofeedback-seriousgame-attentionalstate (accessed on 12 September 2024). The datasets analyzed during this study can be found in the same repository.

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
