# Peer review of "Electroencephalography Neurofeedback Training with Focus on the State of Attention: An Investigation Using Source Localization and Effective Connectivity"

_sensors, 2024, doi:10.3390/s24186056_

Round 1
Reviewer 1 Report
Comments and Suggestions for Authors
Recommendation –Major revisions required
Overall, the authors did a good job in visualizing the different conditions and features they were targeting for this paper. However, the statistical and formal analysis is severely lacking and non-existent in some parts. A lack of a pre-post analysis limits the impact of the NF paradigm. In addition, a lack of trial based analysis and behavior performance also limits the paper’s ability to convince me participants are able to achieve neural modulation.
Major points
- The introduction focuses a lot on ADHD for justification of the attention task and target (theta/beta band) However the actual study recruits healthy participants without symptoms of ADHD. Since this is only done in healthy participants, clinical interpretations should be minimized.
- Can the authors explain why they chose 3 days and only 5 minutes each, is this enough time for NF training?
- The authors should include more details about the task and feedback rate, how fast are speeds updated and updated based on previous N sample points.
- Related to the previous point, can the authors comment on the speed of the source localization algorithm since it is not a particularly fast algorithm for real time implementation.
- The authors say they use SVM to build a classifier of attended/distracted states within the first 10 minuets which is based on a previous paper. In that paper, authors say they use LDA on EEG to reduce the feature set for SVM. A potential point of concern is LDA on EEG and PCA transforms the data into a unique space to distinguish attended/distracted. However, the current paper is using source localized data for feedback. Although they are both using EEG data, the source space and LDA/PCA space are inherently different. The authors should be very explicit in justifying and validating their approach.
- The authors mention they are using a source localized regions for the target signal. Can the authors provide more specifics in which regions/features they are targeting.
- The authors state that temporal lobe is believed to be involved in focusing on a task, it would be interesting to see if activity in these regions is in fact higher in the attended/distracted periods, or if it is higher in the difficult cohort vs easy cohort. It is a bit hard to parse just looking at the full brain maps.
- The authors should include any statistical tests they have done, especially in the connectivity figures since they are not showing all possible connections. How were the ones shown graphically determined.
- The authors should also comment more about the low vs high task differences in results. They are more or less treated as separate analyses in the results and discussion section.
- I encourage the authors to produce more statistical analysis throughout the paper. Some ideas include, any pre-post differences observed, differences in cortical activity between attended vs distracted trials, differences between low and high difficulty groups at each time point (both activity and connectivity measures).
Minor Points
- Introduction line 35/36, change references from singular he/him to plural they because it is not just one person doing the task.
- In data processing 2.3, authors can shorten the threshold rejection paragraph to just say they did Kmeans to find an amplitude threshold consistent with noise and removed previous 500 samples. In addition, if the authors wish to include k means, they should also include how many data points are being used for each run. Is this per trial or using all data recorded so far.
- The authors can consider moving equations to supplemental materials since they are not novel equations and just explain the analysis they are preforming in words.
- Figure 1 looks very blurry, but this might just be a result of PDF conversion.
- Figure 6, Please explain what the different colored bars indicate explicitly.
- Participants provided a written record of experiences but this is not talked about in the results or discussion. Either remove or include this information if relevant.
- “The cortical Regions of Interest (ROI) to generate effective connectivity images were the positions of the electrodes on the scalp, to understand the flow of information between the regions where the signals were collected. Each ROI corresponds to an average of all underlying channels.” This doesn’t make sense, it seems like the authors are saying that the roi are electrodes, but then each ROI is an average of electrodes. Please clarify.
Author Response
Major points
Comments 1: The introduction focuses a lot on ADHD for justification of the attention task and target (theta/beta band) However the actual study recruits healthy participants without symptoms of ADHD. Since this is only done in healthy participants, clinical interpretations should be minimized.
Response 1: Thanks for the suggestion. To minimize it, we changed this: “In [3], children with ADHD in the second session of NF using the TBR protocol, showed decreased theta activity, demonstrating rapid and successful neuroregulation. Another study evaluates and compares the theta/beta ratio in children and adults with ADHD [4]. They demonstrated the possibility to differentiate children with ADHD from the control group by absolute theta values and theta/beta ratio.”
to this: “In [3] they demonstrated the possibility of differentiating children with ADHD from the control group by absolute values of theta and theta/beta ratio.”
Comments 2: Can the authors explain why they chose 3 days and only 5 minutes each, is this enough time for NF training?
Response 2: The number of sessions and the training time were defined based on the literature studied. It was observed that in the first 3 sessions it was already possible to identify differences in the activation of the theta and beta frequencies. And the training duration could not be too long so as not to tire the participants, since the idea was for them to stay focused.
Comments 3: The authors should include more details about the task and feedback rate, how fast are speeds updated and updated based on previous N sample points.
Response 3: Thanks for the suggestion. The following text has been added to topic 2.2. Data collection and NF sessions: “In the neurofeedback training stage, to ensure communication between the user and the feedback given by the system, a Matlab library known as Lab Streaming Layer (LSL) was used. LSL consists of a protocol with libraries and tools that allows the synchronization and transfer of neurophysiological data and biosignals in real time. Then, a matrix buffer was built with 1000 samples per EEG channel and the feedback update rate was based on 500 samples per second.”
Comments 4: Related to the previous point, can the authors comment on the speed of the source localization algorithm since it is not a particularly fast algorithm for real time implementation.
Response 4: The processing of source location and effective connectivity were performed offline. The online stage was the Neurofeedback sessions.
Comments 5: The authors say they use SVM to build a classifier of attended/distracted states within the first 10 minuets which is based on a previous paper. In that paper, authors say they use LDA on EEG to reduce the feature set for SVM. A potential point of concern is LDA on EEG and PCA transforms the data into a unique space to distinguish attended/distracted. However, the current paper is using source localized data for feedback. Although they are both using EEG data, the source space and LDA/PCA space are inherently different. The authors should be very explicit in justifying and validating their approach.
Response 5: We were indeed mistaken. We corrected the text as follows: “This is performed by an algorithm based on Support Vector Machines (SVM) with the Matlab functions fitcsvm() to model the classifier and predict() to predict the class.”
Comments 6: The authors mention they are using a source localized regions for the target signal. Can the authors provide more specifics in which regions/features they are targeting.
Response 6: We correct and specify as follows: “The cortical Regions of Interest (ROI) to generate effective connectivity images were based on Brodmann Areas, with the following areas being manually selected: anterior prefrontal cortex (10), dorsolateral prefrontal cortex (46), prefronta dorsolateral cortex (9), temporal gyrus (20,21,22), somatosensory association cortex (7), associative visual cortex (19) and primary and secondary visual cortex (17,18).”
Comments 7: The authors state that temporal lobe is believed to be involved in focusing on a task, it would be interesting to see if activity in these regions is in fact higher in the attended/distracted periods, or if it is higher in the difficult cohort vs easy cohort. It is a bit hard to parse just looking at the full brain maps.
Response 7: In topic 3.1. Functional Location of Oscillatory Activity for the developed game, we said: "It is possible to identify in the images the regions that showed greater activation (higher normalized power of the cortical current density distribution) for theta and beta frequencies, respectively. The color bar is defined by the maximum absolute value between 0 and 1."
Comments 8: The authors should include any statistical tests they have done, especially in the connectivity figures since they are not showing all possible connections. How were the ones shown graphically determined.
Response 8: The effective connectivity images are showing the connections that are truly significant due to the use of the surrogate data method. And the averages of the users participating in the study are presented in order to find the common connections between the two study groups.
Comments 9: The authors should also comment more about the low vs high task differences in results. They are more or less treated as separate analyses in the results and discussion section.
Response 9: The results and discussion sections have been separated and the results have been expanded.
Comments 10: I encourage the authors to produce more statistical analysis throughout the paper. Some ideas include, any pre-post differences observed, differences in cortical activity between attended vs distracted trials, differences between low and high difficulty groups at each time point (both activity and connectivity measures).
Response 10: The results and discussion section presents a statistical study of the theta and beta frequencies of the three neurofeedback sessions during moments of attention and lack of attention in each study region. In this study, it is possible to analyze the differences in activation in the regions where the images of source localization and effective connectivity were generated.
Minor Points
Comments 11: Introduction line 35/36, change references from singular he/him to plural they because it is not just one person doing the task.
Response 11: We agree with this comment. Therefore, we have changed this: "The information acquired from the EEG is then used by the subject himself so that he can control his own performance." to this: "The information acquired from the EEG is then used by the subjects themselves so that they can control their own performance" (Introduction line 35/36).
Comments 12: In data processing 2.3, authors can shorten the threshold rejection paragraph to just say they did Kmeans to find an amplitude threshold consistent with noise and removed previous 500 samples. In addition, if the authors wish to include k means, they should also include how many data points are being used for each run. Is this per trial or using all data recorded so far.
Response 12: Thanks for the suggestion. We decided to keep the threshold rejection paragraph, since in our opinion it presents important information. But include more information about the k-means used: “In this study, the K-means method was used for every 500 analysis samples. A sliding window of 100 samples and K = 3 (number of clusters) were used. The minimum and maximum limits were defined by the separation of the data into 3 clusters.”
Comments 13: The authors can consider moving equations to supplemental materials since they are not novel equations and just explain the analysis they are preforming in words.
Response 13: Thank you for the suggestion, but we believe it would be interesting to keep the equations together with the text to describe in detail not only in text form but also in mathematical equation form.
Comments 14: Figure 1 looks very blurry, but this might just be a result of PDF conversion.
Response 14: Thanks for the note. We have adjusted the resolution of Figure 1.
Comments 15: Figure 6, Please explain what the different colored bars indicate explicitly.
Response 15: Legends have been added to better describe the boxplot colors. In addition to the text present in lines 350-354.
Comments 16: Participants provided a written record of experiences but this is not talked about in the results or discussion. Either remove or include this information if relevant.
Response 16: This information is not really relevant to the article in question. We have chosen to remove it.
Comments 17: “The cortical Regions of Interest (ROI) to generate effective connectivity images were the positions of the electrodes on the scalp, to understand the flow of information between the regions where the signals were collected. Each ROI corresponds to an average of all underlying channels.” This doesn’t make sense, it seems like the authors are saying that the roi are electrodes, but then each ROI is an average of electrodes. Please clarify.
Response 17: Thank you for pointing this out. This part really got confusing. We changed the text to this: “ The cortical Regions of Interest (ROI) to generate effective connectivity images were based on Brodmann Areas, with the following areas being manually selected: anterior prefrontal cortex (10), dorsolateral prefrontal cortex (46), prefronta dorsolateral cortex (9), temporal gyrus (20,21,22), somatosensory association cortex (7), associative visual cortex (19) and primary and secondary visual cortex (17,18).”
Reviewer 2 Report
Comments and Suggestions for Authors
The paper investigated how neurofeedback helps attention for a task with varying difficulty. Their findings showed that in high-difficulty tasks, brain regions (the medial prefrontal, dorsolateral prefrontal, and temporal areas) were actively involved in maintaining attention. I think the paper needs improvements.
I feel the motivation for selecting TBR is quite shallow. The first paragraph in the introduction mentions some relevant information, but the focus is scattered and incoherent. In addition, some brain regions also were mentioned, but the connection to the present study is unclear. Please expand on the introduction to provide a clear and coherent rationale for selecting TBR and its relevance to the study.
The authors stated that EEG signals were re-referenced to the average reference. However, how this was done is not revealed. Incorrect re-referencing to the average could make the data rank deficient. Depending on the subsequent analysis or circumstances, this issue may affect the results. Either way, the authors should explicitly mention the issue and reveal their re-referencing process.
https://sccn.ucsd.edu/wiki/Makoto's_preprocessing_pipeline#A_study_on_the_ghost_IC_was_published_.28added_on_04.2F04.2F2023.29
The k-means process should be explained in detailed. Did you use sliding windows? What is the input for the clustering? What is the definition of ‘feature’ in k-means here? How did you determine the threshold? How did you decide on the value of ‘k’? and so on. The current description is insufficient.
What method did you use to estimate current source? The paper only mentioned that the method assumed orientation constraints. Please provide a detailed description of the source estimation method used.
I am confused with the ROI in computing connectivity. The authors stated current sources were estimated. Then, why do you have to select electrode positions as ROI? What is the point of estimating cortical sources if the electrode positions are used for connectivity analysis? Did you use any atlas? In addition, list all ROIs.
The section 3 is very chaotic and the focus is scattered. Please move the content about statistical validation to the Methods. Additionally, create a separate Discussion section, following the conventional structure. I find the manuscript combining results and discussion together does not offer any advantages and hinders clarity.
The authors said “Statistical significance is reported using the surrogate data method [52,53].” However, there are no reports of statistical validation in the results. All the current results were not validated.
In addition, the description of statistical tests is quite insufficient. Why do you shuffle “phases”? Why do you perform FT? What do you mean by ‘a time series’? Do you mean a windowing method? If so, you should mention the characteristic of windows. I think the author should explain why the method can validate the results. As permutation tests or generating the surrogated distribution for statistical tests could have a lot of variations, a detailed explanation is needed.
The structure of the paper should be reorganized. Taken together with what I mentioned above, the conclusion is quite long. Some of the contents should be put in the discussion section.
Author Response
Comments 1: I feel the motivation for selecting TBR is quite shallow. The first paragraph in the introduction mentions some relevant information, but the focus is scattered and incoherent. In addition, some brain regions also were mentioned, but the connection to the present study is unclear. Please expand on the introduction to provide a clear and coherent rationale for selecting TBR and its relevance to the study.
Response 1: The following text has been added to the introduction: “Based on the evidence demonstrated on the TBR protocol, it is possible to believe that its analysis can help to find the brain regions related to the attentional state, since in the studies cited it is possible to differentiate attentional states in individuals with and without attentional disorder.”
Comments 2: The authors stated that EEG signals were re-referenced to the average reference. However, how this was done is not revealed. Incorrect re-referencing to the average could make the data rank deficient. Depending on the subsequent analysis or circumstances, this issue may affect the results. Either way, the authors should explicitly mention the issue and reveal their re-referencing process.
https://sccn.ucsd.edu/wiki/Makoto's_preprocessing_pipeline#A_study_on_the_ghost_IC_was_published_.28added_on_04.2F04.2F2023.29
Response 2: Thanks for the note. We have added the reference.
Comments 3: The k-means process should be explained in detailed. Did you use sliding windows? What is the input for the clustering? What is the definition of ‘feature’ in k-means here? How did you determine the threshold? How did you decide on the value of ‘k’? and so on. The current description is insufficient.
Response 3: Thank you for your observation. Details have been added in section 2.3 as follows: “In this study, the K-means method was used for every 500 analysis samples. A sliding window of 100 samples and K = 3 (number of clusters) were used. The minimum and maximum limits were defined by the separation of the data into 3 clusters.”
Comments 4: What method did you use to estimate current source? The paper only mentioned that the method assumed orientation constraints. Please provide a detailed description of the source estimation method used.
Response 4: Section 2.4. Functional localization describes the CCD method used to estimate the current source.
Comments 5: I am confused with the ROI in computing connectivity. The authors stated current sources were estimated. Then, why do you have to select electrode positions as ROI? What is the point of estimating cortical sources if the electrode positions are used for connectivity analysis? Did you use any atlas? In addition, list all ROIs.
Response 5: Thank you for pointing this out. This part really got confusing. We changed the text to this: “ The cortical Regions of Interest (ROI) to generate effective connectivity images were based on Brodmann Areas, with the following areas being manually selected: anterior prefrontal cortex (10), dorsolateral prefrontal cortex (46), prefronta dorsolateral cortex (9), temporal gyrus (20,21,22), somatosensory association cortex (7), associative visual cortex (19) and primary and secondary visual cortex (17,18).”
Comments 6: The section 3 is very chaotic and the focus is scattered. Please move the content about statistical validation to the Methods. Additionally, create a separate Discussion section, following the conventional structure. I find the manuscript combining results and discussion together does not offer any advantages and hinders clarity.
Response 6: Thank you for your feedback and suggestions. We have moved the content on statistical validation to the Methods section. In addition, we have separated the Results and Discussion sections into the conventional structure.
Comments 7: The authors said “Statistical significance is reported using the surrogate data method [52,53].” However, there are no reports of statistical validation in the results. All the current results were not validated.
Response 7: Thank you for your observation, but the generated images represent the maximum absolute values ​​for both the cortical current density distribution and the strength of effective directional connectivity.
Comments 8: In addition, the description of statistical tests is quite insufficient. Why do you shuffle “phases”? Why do you perform FT? What do you mean by ‘a time series’? Do you mean a windowing method? If so, you should mention the characteristic of windows. I think the author should explain why the method can validate the results. As permutation tests or generating the surrogated distribution for statistical tests could have a lot of variations, a detailed explanation is needed.
Response 8: All of the steps presented were used to create a surrogate dataset. We have provided an explanation for the use of each step.
Comments 9: The structure of the paper should be reorganized. Taken together with what I mentioned above, the conclusion is quite long. Some of the contents should be put in the discussion section.
Response 9: The results and discussion sections have been separated. The conclusion was a little long, but only the results obtained were mentioned.
Round 2
Reviewer 1 Report
Comments and Suggestions for Authors
Thanks to the authors for making the revisions. Comments below are aligned with the comment number in the previous revision.
Comment 2: I encourage the authors to include the citations in text for 3 days and 5 minutes of NF training.
Comment 4: Can the authors indicate in the methods which section are used for the real time NF portion and which analysis are done after NF sessions? It seems like NF is done in EEG space based on the SVM model built per person. So the feedback signal is derived from broad band (0.5-100Hz) EEG activity. Then to analyze the effects of NF, source localization is used?
Comment 5: Another follow up question. If every session an svm is built in the first 10 minutes of recording per participant, it is possible that the models will vary day to day for participants (ie lets say SVM model day 1 favors F3 more, then day 2 model it favors O1), then the participant would essentially have two different scalp targets for those two days. If this is the case, then I don’t think source space results are appropriate to discuss since participants could be trained to modulate different parts of the brain per session. Can the authors comment on this point?
Comment 6: Can the authors comment how they chose these regions? Was this done a priori or upon seeing the results of the source localization maps?
Comment 10: I do not see any reference to statistical models on the actual NF session data. Ie: figure 6 and 7. You claim in an earlier point that throughout the three sessions people are able to modulate the signal. However, in figures 6 and 7, it does not seem very convincing that there is session by session difference. The stronger trendline seems to just be attention vs without attention. A repeated measures anova should be run here to show that session 3 theta/beta ratio is higher than session 1 theta/beta ratio. Similarly, in figure 2/3, it is showing differences between low and high difficulty groups in heatmaps. There should be some statistical analysis associated with this low vs high analysis. This could just be in the ROI’s of interest listed out earlier, from Broadmans area map. Adding session by session analysis would strengthen this paper by showing improvement or change over time which is a key aspect of NF training. Without this, what we are seeing could just be task on vs task off since there is no resting state (task off) data to compare to.
Author Response
Comment 2: I encourage the authors to include the citations in text for 3 days and 5 minutes of NF training.
Response 2: We complemented topic 2.2 with the justification for the number of sessions and game time selected. "The number of sessions and game time were defined based on an initial analysis and a time that would not leave the subjects mentally tired, since the game demands a large amount of attention."
Comment 4: Can the authors indicate in the methods which section are used for the real time NF portion and which analysis are done after NF sessions? It seems like NF is done in EEG space based on the SVM model built per person. So the feedback signal is derived from broad band (0.5-100Hz) EEG activity. Then to analyze the effects of NF, source localization is used?
Response 2: Topics 2.2 and 2.3 have been modified to clarify doubts.
Comment 5: Another follow up question. If every session an svm is built in the first 10 minutes of recording per participant, it is possible that the models will vary day to day for participants (ie lets say SVM model day 1 favors F3 more, then day 2 model it favors O1), then the participant would essentially have two different scalp targets for those two days. If this is the case, then I don’t think source space results are appropriate to discuss since participants could be trained to modulate different parts of the brain per session. Can the authors comment on this point?
Response 2: An SVM model built for each session is not a problem since the stimuli given to users are the same in all sessions, forcing participants to activate only regions related to the state of attention. Furthermore, generating models for each session ensures that the actual current mental state of the participants is improved.
Comment 6: Can the authors comment how they chose these regions? Was this done a priori or upon seeing the results of the source localization maps?
Response 2: Thanks for the observation. The following text has been added to topic 2.5: “The selected areas were based on the results of the source localization maps.”
Comment 10: I do not see any reference to statistical models on the actual NF session data. Ie: figure 6 and 7. You claim in an earlier point that throughout the three sessions people are able to modulate the signal. However, in figures 6 and 7, it does not seem very convincing that there is session by session difference. The stronger trendline seems to just be attention vs without attention. A repeated measures anova should be run here to show that session 3 theta/beta ratio is higher than session 1 theta/beta ratio. Similarly, in figure 2/3, it is showing differences between low and high difficulty groups in heatmaps. There should be some statistical analysis associated with this low vs high analysis. This could just be in the ROI’s of interest listed out earlier, from Broadmans area map. Adding session by session analysis would strengthen this paper by showing improvement or change over time which is a key aspect of NF training. Without this, what we are seeing could just be task on vs task off since there is no resting state (task off) data to compare to.
Response 2: Repeated measures anova were performed on the data in figures 2, 3, 6 and 7 and the results are described.
Reviewer 2 Report
Comments and Suggestions for Authors
The paper has shown substantial improvement, but some critical concerns remain.
Comments 2: The authors stated that EEG signals were re-referenced to the average reference. However, how this was done is not revealed. Incorrect re-referencing to the average could make the data rank deficient. Depending on the subsequent analysis or circumstances, this issue may affect the results. Either way, the authors should explicitly mention the issue and reveal their re-referencing process.
https://sccn.ucsd.edu/wiki/Makoto's_preprocessing_pipeline#A_study_on_the_ghost_IC_was_published_.28added_on_04.2F04.2F2023.29
Response 2: Thanks for the note. We have added the reference.
ð Unfortunately, this response is unsatisfactory. It seems the authors did not fully engage with the concern I raised. After reviewing the cited paper, I found it irrelevant to the question at hand. My request pertains to whether the authors retained the initial reference and whether their re-referencing process was properly handled. If further clarification is needed, I encourage you to go to the wiki again and consult relevant literature for a deeper understanding. While many previous studies have overlooked this distinction, it is crucial for avoiding replication issues. To ensure clarity, please explicitly address this concern and clearly explain how the re-referencing process was handled. I formally request the authors to address this issue.
Comments 7: The authors said “Statistical significance is reported using the surrogate data method [52,53].” However, there are no reports of statistical validation in the results. All the current results were not validated.
Response 7: Thank you for your observation, but the generated images represent the maximum absolute values ​​for both the cortical current density distribution and the strength of effective directional connectivity.
ð The response seems to misinterpret my original comment. I was requesting that the authors provide statistical validation for the results, including significant connections or effects. The current description does not include any explicit mention of statistical significance. Without this, the findings cannot be considered properly validated.
Author Response
Comments 2: The authors stated that EEG signals were re-referenced to the average reference. However, how this was done is not revealed. Incorrect re-referencing to the average could make the data rank deficient. Depending on the subsequent analysis or circumstances, this issue may affect the results. Either way, the authors should explicitly mention the issue and reveal their re-referencing process.
https://sccn.ucsd.edu/wiki/Makoto's_preprocessing_pipeline#A_study_on_the_ghost_IC_was_published_.28added_on_04.2F04.2F2023.29
Response 2: Thanks for the note. We have added the reference.
ð Unfortunately, this response is unsatisfactory. It seems the authors did not fully engage with the concern I raised. After reviewing the cited paper, I found it irrelevant to the question at hand. My request pertains to whether the authors retained the initial reference and whether their re-referencing process was properly handled. If further clarification is needed, I encourage you to go to the wiki again and consult relevant literature for a deeper understanding. While many previous studies have overlooked this distinction, it is crucial for avoiding replication issues. To ensure clarity, please explicitly address this concern and clearly explain how the re-referencing process was handled. I formally request the authors to address this issue.
Response 2: We apologize for the misinterpretation. We have added a paragraph explaining the procedure in question: “Then, the signals were filtered with a common average reference (CAR) to reject the spatial common interference. In this filtering, the average signal (interference) of all channels involved is calculated, and this signal is subtracted from each channel, thus eliminating spatial interference.”
Comments 7: The authors said “Statistical significance is reported using the surrogate data method [52,53].” However, there are no reports of statistical validation in the results. All the current results were not validated.
Response 7: Thank you for your observation, but the generated images represent the maximum absolute values ​​for both the cortical current density distribution and the strength of effective directional connectivity.
ð The response seems to misinterpret my original comment. I was requesting that the authors provide statistical validation for the results, including significant connections or effects. The current description does not include any explicit mention of statistical significance. Without this, the findings cannot be considered properly validated.
Response 7: Repeated measures anova were performed on the data in figures 2, 3, 6 and 7 and the results are described.